# Effect of Gamma-Radiation on Zearalenone—Degradation, Cytotoxicity and Estrogenicity

**DOI:** 10.3390/foods9111687

**Published:** 2020-11-18

**Authors:** Thalita Calado, Luís Abrunhosa, Sandra Cabo Verde, Luis Alté, Armando Venâncio, María Luisa Fernández-Cruz

**Affiliations:** 1CEB-Centre of Biological Engineering, University of Minho, Campus de Gualtar, 4710-057 Braga, Portugal; thalita@ceb.uminho.pt (T.C.); luisjap@deb.uminho.pt (L.A.); 2Centro de Ciências e Tecnologias Nucleares (C2TN), Instituto Superior Técnico, Universidade de Lisboa, 1649-004 Lisbon, Portugal; sandracv@ctn.tecnico.ulisboa.pt; 3Departamento de Medio Ambiente y Agronomía, Instituto Nacional de Investigación y Tecnología Agraria y Alimentaria (INIA), Carretera de la Coruña Km 7.5, 28040 Madrid, Spain; luis.alte@inia.es (L.A.); fcruz@inia.es (M.L.F.-C.)

**Keywords:** mycotoxins, *Fusarium* toxins, zearalenone, detoxification, irradiation, toxicity, estrogenicity

## Abstract

Zearalenone (ZEA) is produced in cereals by different species of *Fusarium*, being a non-steroidal estrogenic mycotoxin. Despite having a low acute toxicity, ZEA strongly interferes with estrogen receptors. Gamma-radiation has been investigated to eliminate mycotoxins from food and feed, showing promising results. The present study aims to investigate the gamma-radiation effect on ZEA at different moisture conditions and to evaluate the cytotoxicity and estrogenicity of the irradiated ZEA. Different concentrations of dehydrated ZEA and aqueous solutions of ZEA were exposed to gamma-radiation doses ranging from 0.4 to 8.6 kGy and the mycotoxin concentration determined after exposure by high performance liquid chromatography (HPLC) with fluorescence detection. Following this, the cytotoxicity of irradiated samples was assessed in HepG2 cells, by measuring alterations of metabolic activity, plasma membrane integrity and lysosomal function, and their estrogenicity by measuring luciferase activity in HeLa 9903 cells. Gamma-radiation was found to be effective in reducing ZEA, with significant increases in degradation with increased moisture content. Furthermore, a reduction of cytotoxicity with irradiation was observed. ZEA estrogenicity was also increasingly reduced with increasing radiation doses, but mainly in aqueous solutions. These results suggest reduction of ZEA levels and of its toxicity in food and feed commodities may be achieved by irradiation.

## 1. Introduction

Mycotoxins are toxic fungal metabolites usually found in feeds and foodstuffs [1]. Zearalenone (ZEA) is a non-steroidal estrogenic mycotoxin classified by International Agency for Research on Cancer (IARC) as a group 3 agent [2,3]. This mycotoxin is biosynthesized through a polyketide pathway by several *Fusarium* species, among which *F. cerealis*, *F. crookwellense*, *F. culmorum*, *F. equiseti*, *F. graminearum* (*Gibberella zeae*) and *F. semitectum* are the most relevant ones [4,5,6]. These common soil fungi are found in temperate and warm zones that frequently contaminate worldwide cereal crops, such as barley, maize, oats, rice and sorghum [5,7].

Since ZEA is heat stable, it is not substantially eliminated from raw materials by processing methods commonly adopted in the food industry, thus being detected in many end–products, like bread or breakfast cereals [8,9]. This constitutes a serious health problem because ZEA has important toxicological properties. The most relevant one is its estrogenicity, since several studies demonstrate that ZEA is implicated in reproductive disorders that result in functional and morphological alterations in reproductive organs in different domestic animal species, particularly in pigs [10,11,12]. After oral administration, ZEA is rapidly absorbed and biotransformed, mainly in the liver, to α-Zearalenol (α-ZOL) and β-Zearalenol (β-ZOL), then reduced to α-Zearalanal (α-ZAL) and β-Zearalenal (β-ZAL), respectively [13,14]. ZEA, α-ZAL and β-ZAL, catalyzed by uridine diphosphate glucuronyl transferases, can be conjugated with glucoronic acid which facilitates their elimination via urine, feces and bile [4,5]. Nonetheless, conjugated metabolites excreted via the bile are reabsorbed by the intestinal mucosal cells, ultimately entering again in the liver and the systemic circulation via the portal blood supply [5,15]. This circulation of ZEA and its derivatives extends their biological half-life and increases their total toxicity. Due to the structural similarity of ZEA and its metabolites with endogenous estrogen (17β-estradiol (E2)), these compounds can bind to estrogen receptors [16], causing reproductive disorders. In addition to estrogenic effects, the toxicity of ZEA includes genotoxicity [17], cytotoxicity [18,19] immunotoxicity [18,20], reproductive toxicity [11,21], increase of reactive oxygen species in cells [21,22], developmental toxicity [23], hematotoxicity and hepatotoxicity [24]. Thus, strategies to reduce or eliminate the toxic effects of ZEA are needed to improve food safety and to minimize economic losses in livestock production [25].

One physical method that can be used to control the presence of mycotoxins in food and feed is irradiation. Presently, due to the power of penetration of gamma rays along with its broad-spectrum efficacy against microorganisms, gamma-radiation is the preferred method to irradiate commodities [26]. In the specific subject of mycotoxins, gamma-radiation can, on the one hand, inhibit or delay the development of the mycotoxigenic fungi and, consequently, the production of mycotoxins; and on the other hand, exerts a direct degradation action on mycotoxins [27]. The elimination of mycotoxins by gamma-radiation is a subject that has been widely investigated but the available literature is not always consensual about its efficacy. Many factors may influence the irradiation process, such as the absorbed dose, the average dose rate, the initial moisture content and mycotoxin concentration [27,28,29], explaining some contradictory results reported in the literature. Moreover, differences in the degradation products formed have also been reported [28].

The degradation of mycotoxins may form other compounds (radiolytic products) that can be as or more toxic than the original mycotoxin. According to Rychlik et al. [30], the potential exposure to modified mycotoxins is an additional risk to human and animal health. Thus, the study of mycotoxins degradation should be accompanied with assays on the safety of the irradiated product. The detection, identification and isolation of each radiolytic product is the ideal option. However, due to the diversity of the radiolytic products that may be produced and to their very low concentration, this is not always feasible [28]. One excellent alternative is to study the toxicity of radiolytic product as a whole using in vitro assays.

Using cells in in vitro assays has two main advantages: to minimize the animal use and to allow a wider range of chemicals and concentrations to be tested [31,32]. Nevertheless, sensitive and rapid cell viability assays are required to use cells in toxicity tests. The use of fluorescent dyes in microwell plates viability tests have several advantages: several indicator dyes are commercially available which allows a large range of cellular parameters to be monitored [33]. Alamar Blue (AB) to detect changes in energy metabolism, carboxyfluorescein diacetate acetoxymethyl ester (CFDA-AM) to evaluate membrane integrity, and neutral red (NR) to evaluate lysosomal function are examples of indicator dyes [34].

According to previous studies, ZEA can exert liver toxicity [5,35,36]. Maaroufi et al. [37] concluded that ZEA exerts liver toxicity in rats and Čonková et al. [38] observed the same toxicity in rabbits. The human hepatoma HepG2 cells were reported to preserve many of the properties of primary liver cells and, for this reason, this line of cells can be a good approximation to a real situation [32,39,40].

The estrogenicity of ZEA after irradiation should also be studied to prove that the radiolytic products do not increase the estrogenic potency. According to the Organization for Economic Co-operation and Development (OECD), one cell line that can be used to screen and test potential endocrine disrupting chemicals is the HeLa cells transfected to express the human estrogen receptor alpha (hERα)–the hERα-HeLA-9903 [41].This cell line also presents a luciferase reporter gene. When the estrogenic compound is linked to hERα there is an increase of cellular expression of the luciferase enzyme. In the presence of luciferin the increase of bioluminescence can be measured with a luminometer.

The purposes of this study were (i) to investigate the effect of different doses of gamma-radiation on the degradation of ZEA under different moisture conditions; (ii) to evaluate the cytotoxicity of irradiated ZEA; and (iii) to evaluate estrogenicity of irradiated ZEA.

## 2. Materials and Methods

### 2.1. Chemicals and Reagents

Standard of ZEA (Z2125-10MG, Sigma) was obtained from Sigma-Aldrich (Sintra, PT). Ethanol was obtained from Panreac (Barcelona, ES) and methanol from Merck (Lisbon, PT). Fetal bovine serum (FBS), cell culture EMEM (Eagle’s Minimum Essential Medium), ultraglutamine 1 (L-Gln) (200 μmol L^−1^), non-essential amino acids (NEAA) 100X, Trypsin-Ethylenediamine tetraacetic acid (Trypsin-EDTA) (200 mg L^−1^ EDTA, 17,000 U trypsin L^−1^), penicillin and streptomycin (P/S) (10,000 U mL^−1^/10 mg mL^−1^) were purchased from Lonza (Barcelona, ES). Charcoal-dextran stripped fetal bovine serum (FBS-charcoal), Phenol red-free Minimum Essential Medium (MEM) and kanamycin (Kan) were sourced by from PAN-Biotech (Aidenbach, DE). AlamarBlue, 5-carboxyfluorescein diacetate and acetoxy methyl ester (CFDA-AM) were obtained from Life Technologies (Madrid, ES). Sodium dodecyl sulfate (SDS), neutral red (3-amino-7-dimethylamino-2-methylphenanzine hydrochloride) solution (0.33%), 17β-estradiol (E2), glacial acetic acid, in vitro Toxicology Assay Kit Resazurin based, and dimethyl sulfoxide (DMSO) were acquired from Sigma-Aldrich (Madrid, ES). High grade purity water (>18 MΩ/cm) from a Milli-Q Element A10 Century water purification system (Millipore Iberia, ES) was utilized.

### 2.2. Preparation of ZEA Solutions and Irradiation Process

A stock solution of ZEA at a concentration of 1 mg mL^−1^ was prepared in 10 mL of methanol using commercial standard of ZEA and stored at −20 °C until use. To prepare the ZEA samples of 3 μmol L^−1^ (0.955 μg mL^−1^), the proper amount of the stock was pipetted into clean amber 2 mL vials. Samples were then concentrated to dryness under a nitrogen gentle stream using a sample concentrator (50 °C). For studies with different moisture contents, three sets of ZEA samples were prepared. A set of samples was prepared in deionized water (H_2_O_dd_), another set was prepared in water/methanol (50/50, *v*/*v*) and the last set was kept dry. Samples were then preserved in amber vials and stored at −20 °C until irradiated. For cytotoxicity and estrogenicity evaluation, ZEA samples at concentrations of 60 µmol L^−1^ (19.1 μg mL^−1^) in H_2_O_dd_ or dried were prepared as described above. The higher ZEA concentration was required for cytotoxicity detection.

The irradiations were carried out in a Co–60 experimental equipment Precisa 22 (Graviner Manufacturing Company Ltd., London, UK) with four sources and a total activity of 177 TBq (4.78 kCi; February 2014), situated at C2TN, at room temperature. The average dose rate was previously determined by Frick reference dosimeter and was 1.8 kGy h^−1^ [42]. To monitor the irradiation process, estimation of the highest and the lowest dose absorbed by samples, two routine dosimeters were used (Amber Perspex dosimeters, Batch X, Harwell Company, UK). In order to estimate the dose, the thickness and absorbance of Amber Perspex dosimeters were measured, in a micrometer (Mitutoyo America Corporation, Aurora, IL, USA) and in a Ultraviolet–visible Spectrophotometer (UV-VIS Spectrophotometer) (UV 1800, Shimadzu, Vernon Hills, IL, USA) at 603 nm respectively, according to a previous calibration curve. Samples of ZEA (dehydrated, methanol:water solution, and in water) were irradiated at 0.4, 0.9, 1.7, 2.5, 5.4 and 8.6 kGy, in triplicates. To evaluate the cytotoxicity and estrogenicity after irradiation, only dried and aqueous solutions of ZEA at 60 µmol L^−1^ were submitted to radiation doses of 2.4 and 10.3 kGy. For each condition, non-irradiated controls were also prepared.

### 2.3. Determination of ZEA Levels

The high performance liquid chromatography (HPLC) analysis was adapted from Keller et al. [43]. The chromatographic apparatus consisted of a Varian Prostar 210 pump, a Varian Prostar 410 autosampler, a Jasco FP-920 fluorescence detector set at λexc = 280 nm and λem = 460 nm, a Varian 850-MIB (Modular Interface Box) data system interface and a Galaxie chromatography data system. The separation was achieved with a C18 reversed-phase YMC-Pack ODS-AQ analytical column (250 × 4.6 mm I.D., 5 μm), fitted with a pre-column of the same stationary phase, and a 25 min isocratic run. The column temperature was set to 30 °C. The mobile phase was prepared with methanol, water and acetic acid (65:35:1, *v*/*v*/*v*), was filtered through a 0.2 μm membrane filter (GHP (hydrophilic polypropylene), Gelman) and degassed by sonication. The flow rate was 1.0 mL min^−1^ and the injection volume was 50 μL. ZEA was recognized by retention times (21 min) and quantified by measuring peak areas and comparing them with a calibration curve. Two calibration curves, in mobile phase, were prepared. The first one, with ZEA concentrations from 0.8 µmol L^−1^ to 3.1 µmol L^−1^, was used with the detector gain set to 1000 to quantify the samples of ZEA with 3 µmol L^−1^. The second one, with ZEA concentrations from 3.9 µmol L^−1^ to 63 µmol L^−1^, was used with gain set to 100 to quantify 60 µmol L^−1^ ZEA samples.

### 2.4. Cytotoxicity Studies

#### 2.4.1. Cells Culture and Exposure

As mentioned above, for the cytotoxicity study, ZEA samples at concentrations of 60 µmol L^−1^ in H_2_O_dd_ or dried were prepared. The cell line Hep G2 was acquired from the American Type Culture Collection (ATTC) (Manassas, VA, USA). These cells were cultured in 75 cm^2^ Cell Star Cell Culture flasks (Greiner Bio-One GmbH, Frickenhausen, DE) in EMEM supplemented with 10% FBS, 1% NEAA, 1% L-Gln and 1% P/S (from now referred as EMEM+), and incubated at 37 °C in a humidified 5% CO_2_ atmosphere. Twice a week, there was a flask split using phosphate-buffered saline (PBS)/EDTA and trypsin.

The cytotoxicity assays parameters defined in this study were based on previous results to ochratoxin A [44]. Briefly, a Hep G2 cell suspension in EMEM+ (50 × 10^5^ cells mL^−1^) was prepared. One hundred microlitres of this suspension were seeded into each well of transparent, flat-bottom 96-well plates (Greiner Bio-One GmbH, Frickenhausen, DE). The plates were incubated for 24 h and exposed for 48 h to different concentrations of ZEA. Exposure concentrations of irradiated and non-irradiated ZEA samples were prepared by drying (if suspended in water) as described in Section 2.2 and re-suspending in EMEM+ supplemented with 0.5% of DMSO (improves solubility) to a final concentration of 60 µmol L^−1^. Then, those solutions were applied to the cell culture plate, where successive serial dilutions were performed (dilution factor: 2). Increasing concentrations of SDS (15.6–500 µmol L^−1^, dilution factor: 2/3) were used to treat a subset of wells, to serve as positive control. As negative control, cells were treated with EMEM+, while cells treated with 0.5% (*v*/*v*) DMSO/EMEM+ were the vehicle control.

#### 2.4.2. AB, CFDA-AM and NR Uptake (NRU) Assays

The AB, CFDA-AM and NRU assays were carried out according the methodology described by Calado et al. [44] and Lammel et al. [45], using with the same set of cells. Briefly, after the exposure medium was removed, cells were washed twice with 200 μL of phosphate-buffered saline (PBS). Then, to each well, 100 μL phenol red-free MEM containing 1.25% (*v*/*v*) AB and 4 µmol L^−1^ CFDA-AM were added, and plates were incubated for 30 min in the dark as described earlier. After incubation, for the AB assay, the fluorescence intensity was measured at λexc = 532 nm and λem = 590 nm using a microplate reader (Tecan Genios, Tecan Group Ltd., Männedorf, Switzerland). Subsequently, for the CFDA-AM assay, the fluorescence intensity was measured at λexc = 485 nm and λem = 535 nm. The medium was removed after these reading, and cells were rinsed once with PBS. For NRU assays, a NR solution (0.03 mg mL^−1^ in phenol red-free MEM) was prepared, 100 μL of this NR solution were added to each well and these plates were incubated again for 1 h in the dark. After incubation, the NR solution was removed, the cells were washed twice with 200 μL PBS, and the retained NR in the cells was extracted with an acidified solution composed of 1% glacial acetic acid and 50% ethanol in Milli-Q water (150 μL/well). NR fluorescence was measured at λexc = 532 nm and λem = 680 nm. To all assays, the florescence values were corrected for the cell-free control and normalized against the vehicle control. Fluorescence spectra of ZEA samples (irradiated or not) did not exhibit significant fluorescence at the wavelengths of excitation and emission that were used in the AB, CFDA-AM and NRU.

### 2.5. Estrogenicity Studies

#### 2.5.1. Cell Culture and Exposure

To conduct estrogenicity studies, the hERα-HeLa-9903 cell line has been used. This cell line is derived from a human cervical tumor, and has two stably inserted constructs: (i) the hER expression construct (encoding the full-length human receptor), and (ii) a firefly luciferase reporter construct of an Estrogen-Responsive Element (ERE). The cell line hERα-HeLa-9903 was obtained from the Japanese Collection of Research Bioresources (JCRB) Cell Bank (Osaka, Japan). It was cultured in 75 cm^2^ cell culture flasks treated by vacuum Gas plasma (Becton Dickinson, France) in phenol red-free MEM supplemented with 10% FBS-charcoal, 1% L-Gln, 1% Kan and 1% P/S (in the following text referred to as MEM+). The flasks were incubated at 37 °C in a humidified 5% CO_2_ atmosphere and split twice a week using PBS/EDTA and trypsin. A Hela 9903 cell suspension (10 × 10^5^ cells mL^−1^) in MEM+ was prepared. Hela 9903 cells were seeded into opaque, flat-bottom 96-well plates (Perkin Elmer, Frickenhausen, DE), by adding 100 μL of cell suspension to each well and were incubated as described above for 3 h.

For the determination of ZEA estrogenicity, irradiated and not irradiated ZEA samples, the ZEA solutions of 3 and 60 µmol L^−1^ were diluted into MEM+ to achieve a work solution of 1 µmol L^−1^. The ZEA working solutions were applied to the cell culture plate in which successive serial dilutions were performed (by a factor of 2). As a positive control, a subset of wells was treated with increasing concentrations of E2 (0.004 nmol L^−1^–0.125 nmol L^−1^, dilution factor 2), while cells treated with MEM+ served as negative control and vehicle control. These plates were incubated in a humidified CO_2_ atmosphere, at 37 °C, for 24 h, and were then subjected to analysis.

#### 2.5.2. Transactivation Assay

hERα-HeLa-9903 cell line allows for conducting the transactivation assay. The assay is used to signal the binding between the estrogen receptor and a ligand. Following ligand binding, the receptor-ligand complex translocates to the nucleus, binds to specific DNA response elements (ERE) and transactivates a firefly luciferase reporter gene. This will result in increased cellular expression of luciferase enzyme. The luciferase enzyme transforms its substrate, luciferin, to a bioluminescence product that can be quantitatively measured with a luminometer. The assay was conducted following a modified version of the OECD Guidelines 455 [41]. Firstly, the viability of cells was confirmed by adding 5 μL of resazurin solution per well and by incubating plates for 90 min in the dark at 37 °C in a humidified 5% CO_2_ atmosphere. After the incubation period, the fluorescence intensity was measured at λexc = 532 nm and λem = 590 nm. The results of this assay are not included as no toxic effects were observed at the doses tested indicating that the assay has been conducted adequately. Luciferase activity was then measured using a luciferase reporter gene assay kit (Biodetection Systems, Amsterdam, NL) according to the manufacturer’s instructions with small modifications. Briefly, 90 μL of PBS (pH 7.5) and 30 μL of the lysis buffer were added. After 15 min, 80 μL of the luciferase reagent was added and luminescence was, immediately, measured using a liquid scintillation counter (1450 MicroBeta Trilux, PerkinElmer, Spain). The luminescence values were normalized against the negative control.

### 2.6. Statistical Analysis

Sigma Plot version 12.0 (Jandel Scientific, San Rafael, CA, USA) was used for statistical analysis. Significant differences between ZEA concentrations in non-irradiated and irradiated samples were evaluated, and means were compared by analysis of variance, followed by Duncan’s post-test. Results of cytotoxicity and estrogenicity assays are represented by the means and the corresponding standard errors (SEM) of at least three independent experiments, performed in triplicate. Significant differences among treatments were determined by one-way repeated measures analysis of variance (rmANOVA, *p* < 0.05, α 0.05). Normality (Shapiro-Wilk test, *p* < 0.05) and equal variance (*p* < 0.05) were tested beforehand. Significant differences between treatments and the controls were determined by applying Dunnett’s Post hoc test to one-way ANOVA analyses (*p* < 0.05, *p* < 0.01 and *p* < 0.001).

## 3. Results and Discussion

### 3.1. ZEA Concentration after Irradiation

The effect of gamma-radiation doses on ZEA samples at the lower concentration tested (3 µmol L^−1^) are presented in Figure 1. The maximum elimination was observed in samples of ZEA dissolved in water. In this condition, the lowest radiation dose applied (0.4 kGy) was sufficient to complete the total degradation of ZEA. On the opposite side, the irradiation of dried ZEA presented the lowest reductions in ZEA concentration. In this case, significant reductions of ZEA (between 65% and 87%) were only observed with doses ≥1.7 kGy and the complete elimination of ZEA was not achieved even at the highest dose tested (8.6 kGy). In the assays performed with the mycotoxin dissolved in water/methanol solution, a significant ZEA reduction of 24% was observed at 0.9 kGy, while a reduction of 97% was observed for the highest dose of radiation. The HPLC analysis did not revealed fluorescent degradation products that may have been produced during the irradiation process of ZEA. Major ZEA derivatives, α-zearalenol (α-ZOL), β-zearalenol, (β-ZOL), zearalanone (ZAN), α-zearalanol (α-ZAL) and β-zearalanol (β-ZAL), were not detected. As mentioned previously, the effectiveness of the irradiation process can be affected by several factors, making a comparison with studies available in the literature difficult. Nevertheless, the observed increase of ZEA degradation with increasing gamma-radiation doses (Figure 1) corroborates with some published works. For example, Hooshmand and Klopfenstein [46] demonstrated a 25% reduction of ZEA concentration after irradiation of corn at 10.0 kGy and Aziz et al. [9] verified a total elimination of ZEA in wheat and flour exposed to a radiation dose of 8.0 kGy.

The effect of moisture is overt in our results, particularly when low radiation doses were used. This observation is in agreement with studies using ZEA [47] and using other mycotoxins [48,49,50]. The higher effect of radiation with water is justified by its radiolysis, in which water ionization occurs. In this process, the splitting of water molecules occurs into positively charged radicals (H_2_O^+^) and negative free solvated electrons (e−); which after various recombination and cross-combination reactions originates the reactive species e−aq, H•, HO•, HO_2_•, OH−, H_3_O+, H_2_, and H_2_O_2_ [51]. These compounds/radicals react with double bonds, as in the ones found in aromatic or heterocyclic rings [48], and may explain the higher reductions observed for ZEA. Eventually, these mechanisms may reduce the mutagenicity and toxicity of mycotoxins.

On the other hand, ZEA elimination in 60 µmol L^−1^ samples was less efficient than previously observed in the 3 µmol L^−1^ ZEA samples. As mentioned above, for 3 µmol L^−1^ aqueous samples the complete elimination of ZEA was observed at very low radiation doses (≥0.4 kGy). However, for 60 µmol L^−1^ aqueous samples, a significant reduction of ZEA was only achieved at higher doses (90% and 96% with 2.4 kGy and 10.3 kGy, respectively). Concerning the irradiation of 60 µmol L^−1^ dried ZEA samples, with 2.4 kGy and 10.3 kGy reductions of 18% and 21% were only obtained, respectively. Calado et al. [27] mentioned that the mycotoxin concentration is one of the aspects involved in effectiveness of mycotoxin irradiation process. Van Dyck and others [52] verified that the effect of gamma-radiation was substantially reduced when the concentration of aflatoxin B1 (AFB1) was increased 50 times. Similar results were reported by Mutluer and Erkoc [50] for aflatoxins (B1, B2, G1 and G2), and by Kalagatur and collaborators [53] for ZEA. In spite of this, other authors have reported the opposite effect. Jalili et al. [49], Zhang et al. [54] and Abdel-Rahman et al. [55] described that an increase of AFls concentration seems to increase the irradiation effectiveness in samples.

### 3.2. Cytotoxicity of Irradiated ZEA

The removal of mycotoxins from food and feed is an area that has gathered research interests and gamma-radiation processes has shown promising results. However, the confirmation of the decrease of toxicity after mycotoxins irradiation has been poorly studied.

The cytotoxic effect of ZEA on Hep G2 cells, measured through AB, after 48 h of incubation, is shown in Figure 2a,b. As mentioned above, the cytotoxicity assays were performed with serial half-dilutions of the 60 µmol L^−1^ irradiated dried and water-dissolved ZEA samples. Non-irradiated ZEA samples at concentrations between 0.5 to 60 µmol L^−1^ were also tested causing a significant decrease of Hep G2 cells viability that achieved around 60% at the highest concentration (Figure 2a,b).

Alamar Blue is a commercial preparation of the dye resazurin, which is converted by viable cells to a fluorescent form. The diminishment of fluorescence indicates an impairment of cellular metabolism [33]. Comparing with controls, non-irradiated ZEA samples showed a significant reduction of fluorescence both in water and dried form. Regarding non-irradiated ZEA in water, although the trend was discernible at concentrations as low as 0.9 µmol L^−1^, the first statistical significant difference, respecting the vehicle control, was only detected at 15 µmol L^−1^ (Figure 2a). Also, for dry non-irradiated ZEA, the trend was perceptible at concentrations of 0.5 µmol L^−1^, but the first statistical significant difference regarding the vehicle control was also only detected at 15 µmol L^−1^ (Figure 2b).

ZEA samples irradiated at 2.4 kGy in the presence of water (Figure 2a), showed a significant statistical reduction of the toxic effect observed with non-irradiated. This result indicates that this radiation dose is effective in eliminating ZEA and its associated toxic effects for ZEA concentrations below 30 µmol L^−1^. At the radiation dose of 10.3 kGy the toxic effect of ZEA were eliminated even at the higher ZEA concentration.

For dehydrated samples a different trend was observed (Figure 2b). In this case, the pattern observed with non-irradiated and with irradiated samples was very similar. This result indicates that the reduction in toxicity after irradiation was low and similar for both doses of radiation. It is necessary to remember that, under these conditions, ZEA reductions of 18% and 21%, for 2.4 kGy and 10.3 kGy were observed, respectively. In other words, the real ZEA concentration in samples was close and, thus a similar effect was obtained.

Figure 3a,b show the cytotoxic effect of ZEA measured through the CFDA-AM assay. This assay is based on the conversion by cytosolic esterases of CFDA-AM to its fluorescent product 5-CF, which occurs only in cells with intact plasma membrane, that are able to retain the esterases [34]. For samples of ZEA in water, only the higher concentration of the non-irradiated sample presented significant statistical differences in respect to the vehicle control. Similar results were obtained for ZEA dry samples. With this reagent/dye, the decrease in fluorescence intensity was less prominent than with AB. This suggested that ZEA has a higher effect on the metabolic activity of the cell than in the plasma membrane activity. The same effect was observed by Ayed-Boussema et al. [2], which verify that ZEA reduced HepG2 cells proliferation (IC_50_ about 100 µmol L^−1^) and concluded that the reduction of cells is mainly due to apoptosis rather than necrosis only observed at high concentrations. These authors concluded that apoptosis is the major cause of ZEA-induced cells death and suggested that ZEA induces an activation of pro-apoptotic genes and does not interfere in anti-apoptotic proteins, causing an imbalance that leads to apoptosis. Furthermore, proteins and genes activated by ZEA lead to the disruption of mitochondrial membrane [2], explaining the decrease of fluorescence when AB was used. With CFDA-AM, ZEA toxicity (dried and dissolved in water) was only reduced after irradiation with the highest dose (10.3 kGy) (Figure 3a,b). In relation to non-irradiated ZEA, CFDA-AM fluorescence was increased respectively by 41% and 27% for the dried and dissolved conditions, showing significant reduction of toxicity.

Figure 4a,b show the cytotoxic effect of ZEA on Hep G2 cells, measured through NR, after incubation (48 h). The NRU assay is based on the accumulation of NR in functional lysosomes. In this case, the trend was similar to the AB assay for aqueous and dehydrated samples. With NR, ZEA toxicity of samples dissolved in water was reduced after irradiation. At the 60 µmol L^−1^ ZEA concentration level, an increase of NR fluorescence of 23% and 53% was observed with 2.4 and 10.3 kGy, respectively. In all these assays, the toxicity remaining in samples after irradiation is probably due to the presence of non-degraded ZEA rather than to the degradation products. No toxigenic compounds were generated after irradiation because no increase of toxicity was observed.

As mentioned above, ZEA detoxification by gamma irradiation is poorly studied and as a result there are few studies about the toxicological safety of degradation products from irradiated ZEA. Kalagatur and colleagues [53] tested in macrophage cell line (RAW 264.7) the toxicity of irradiated and non-irradiated aqueous ZEA solutions. Results of MTT (3-(4,5-dimethylthiazol-2-yl)-2,5-diphenyltetrazolium bromide)assay and live/dead cells assay showed significant reductions of toxicity. These results corroborate findings from the current work.

### 3.3. Estrogenicity of Irradiated ZEA

The estrogenic effect of ZEA and its metabolites are known and largely studied by several authors [10,12,16]. The ideal detoxification method must decrease the estrogenicity of ZEA and of eventual degradation products. The dose-response curves shown in Figure 5a were obtained after exposure of hERα-HeLa-9903 cells for 24 h to irradiated ZEA aqueous samples with initial concentration of 3 µmol L^−1^. Results reveal that the exposure to non-irradiated ZEA for 24 h produce an increase of the luminescence intensity, indicating an intensification of luciferase activity. This rise was discernible at concentrations as low as 6.1 × 10^−5^ µmol L^−1^, but the first statistical significant difference, respecting the vehicle control, was only detected at 1.95 × 10^−3^ µmol L^−1^. All the irradiated sample doses showed no significant difference when compared to the vehicle control between them. This result shows that the two radiation doses tested in these conditions make a complete reduction of ZEA estrogenicity. These results corroborate with the HPLC results where no ZEA was detected (Figure 1). When the irradiation was made in dehydrated conditions (Figure 5b) the result was very different. In this case, only a small reduction of the luminescence intensity was observed. Once again, the results agree with HPLC results. In samples where the total destruction of ZEA by radiation was not observed, the estrogenicity was not reduced. In spite of this, no increase of estrogenicity with irradiation was verified.

The effect of irradiation on estrogenicity of ZEA samples with initial concentration of 60 µmol L^−1^ is presented in Figure 6a,b, respectively. The irradiation of this high dose was not sufficient to eliminate ZEA estrogenicity in aqueous samples (Figure 6a) although it was adequate to reduce the cytotoxicity (Figure 2a). After irradiation with the two doses tested, dehydrated ZEA samples with initial concentration of 60 µmol L^−1^ (Figure 6b) presented the same trend as non-irradiated samples. As mentioned above, the reduction of ZEA concentration on these samples was only 20%. Thus, the estrogenicity observed in irradiated samples was due to the remaining ZEA. Once again, the results point out the significant impact of water during the radiation process to an efficient detoxification of ZEA.

The current Commission Regulation (EC) No 1881/2006 [56] set the maximum ZEA levels allowed in different foods. The higher level is 400 µg kg^−1^ for refined maize oil. This concentration is equivalent to approximately 1 µmol L^−1^ and it is below the lower concentration tested (3 µmol L^−1^) which the irradiation process was able to destroy. Therefore, within this concentration range, gamma-radiation can potentially destroy a substantial amount of ZEA and eliminate its toxicity in food and feed commodities, increasing their safety. However, the moisture content in food matrices is determinant in order to ensure the efficiency of irradiation process. Kalagatur et al. [53] studied the efficiency of irradiation on ZEA detoxification from fruit juices (orange, pineapple, and tomato) and verified that the reduction levels in these matrices were similar to the reduction of ZEA in water. These authors concluded that irradiation could be an efficient post-harvest food processing technique for ZEA detoxification from fruit juices, since, despite the high dose of 10 kGy, has minimal effect on the quality of fruit juices such as: decrease of antioxidant activity, small changes in sensory attributes and acidity increased.

ZEA is most commonly found in cereals, and their typical moisture content is below 15%. So, the direct use of irradiation on grains may not destroy substantial levels of ZEA, especially if concentrations are very high. Nonetheless, the efficiency of irradiation may be increased if this technology is incorporated in some steps of well-established grain processing methods such as wet milling. This process involves a step where grain is soaked in water wherein the radiation can be applied. Recently, Sebaei and collaborators [47] studied detoxification efficiency of irradiation on food grains (wheat, white corn and yellow corn) and verified that wheat, which is lower in fat and higher in water, showed the highest reduction level, while corn, with a higher fat content and lower water content, showed a lower reduction level. More research is needed on food samples naturally contaminated with ZEA in order to conclude about the applicability of this method of detoxification. The organoleptic and nutritional proprieties are also important points to investigate in order to evaluate if gamma-radiation may affect important food characteristics. As it happens with others decontamination methods, it is necessary to see which foods are likely to be exposed to gamma-radiation without compromising its properties.

## 4. Conclusions

In this research, gamma irradiation process was demonstrated to be a useful treatment for reducing ZEA levels in vitro. However, the water content during the irradiation process, such as that observed by other researchers for other mycotoxins, play a crucial role on the efficiency of ZEA degradation. With respect to the cytotoxicity of ZEA radiolytic products, the results were promising, since a decrease in ZEA concentration after irradiation was always followed by a reduction in toxicity, especially when ZEA is in contact with water. The same results were observed for the estrogenic activity of irradiated ZEA. These results can indicate that irradiation process applied for ZEA detoxification can be a safe method to reduce ZEA health associated risk. More studies are needed about cytotoxicity and estrogenicity ZEA reduction with gamma irradiation in naturally contaminated food.

## Figures and Tables

**Figure 1 foods-09-01687-f001:**
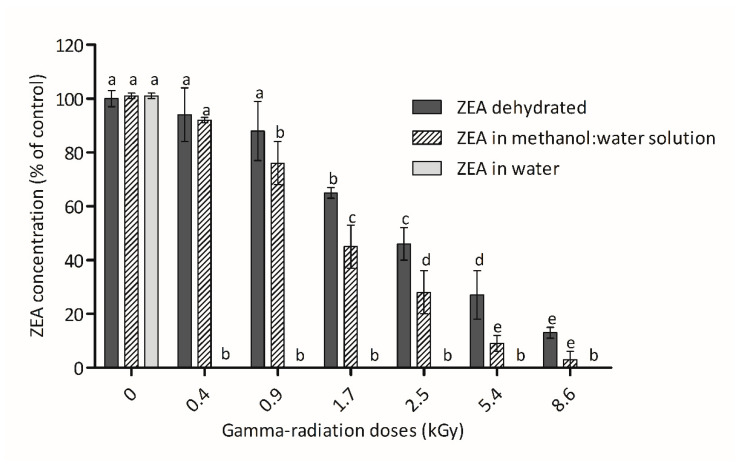
Effect of increasing gamma-radiation dose on Zearalenone (ZEA) (3 µmol L^−1^) in different conditions. Significant differences in each condition (dehydrated, methanol:water solution, and in water) are indicated with different letters, by analysis of variance (one-way rmANOVA, Duncan’s Post-hoc test, *p* < 0.05).

**Figure 2 foods-09-01687-f002:**
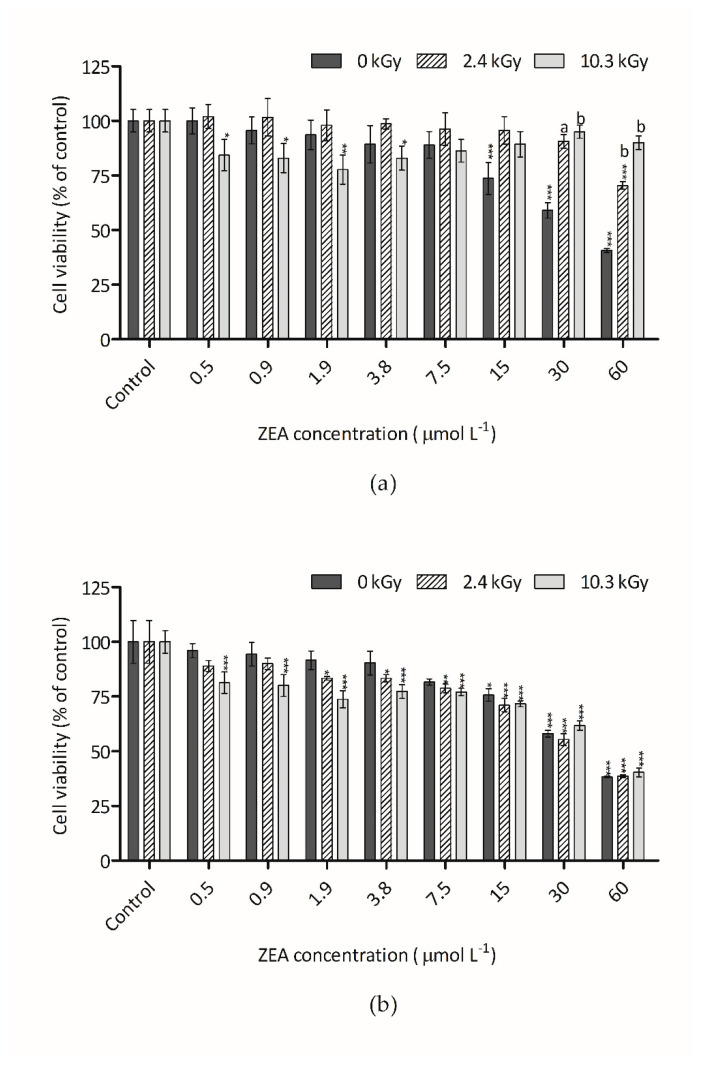
Effect of ZEA samples on HepG2 cell viability assessed by the Alamar Blue assay: (**a**) Irradiated and non-irradiated water-dissolved ZEA samples; (**b**) Irradiated and non-irradiated dried ZEA samples. Bars represent the mean and their standard error (SEM) of at least three independent repetitions. Significant differences with respect to the vehicle control (one-way rmANOVA, Dunnett’s Post-hoc test) are indicated as follows: * (*p* < 0.05), ** (*p* < 0.01), *** (*p* < 0.001). Statistically significant differences with respect to non-irradiated sample (0 kGy) and the irradiated samples (2.4 kGy and 10.3 kGy) (one-way rmANOVA, Dunnett’s Post-hoc test) are indicated as follows: a (*p* < 0.01), b (*p* < 0.001), no letter (no statistical differences).

**Figure 3 foods-09-01687-f003:**
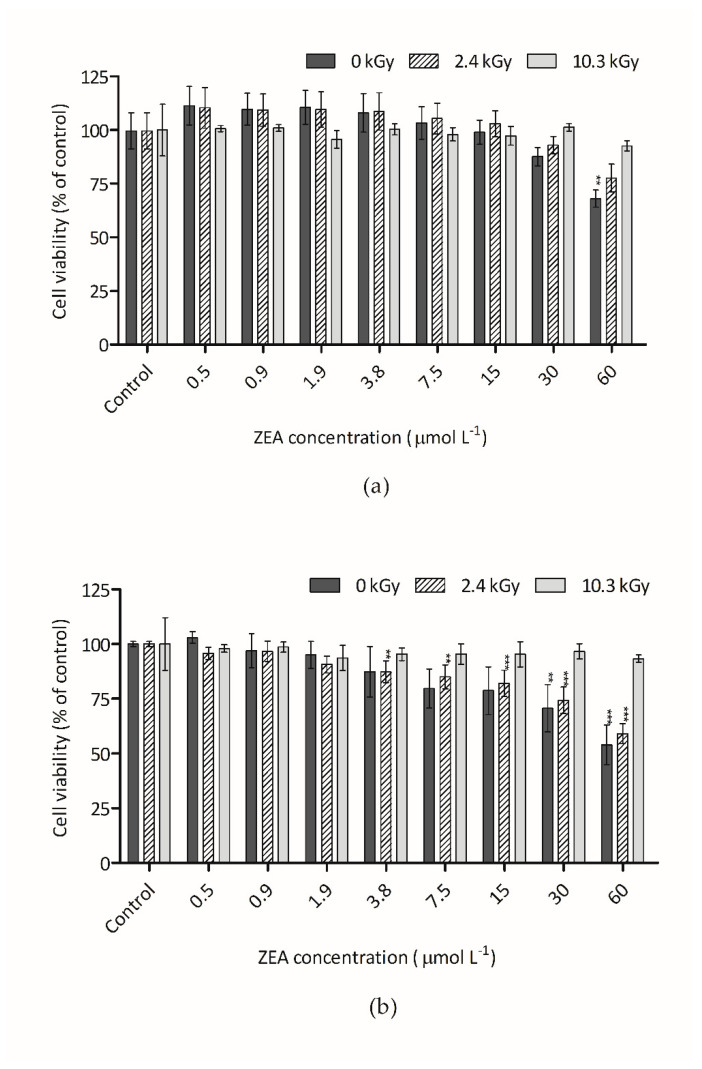
Effect of ZEA samples on HepG2 cell viability assessed by the CFDA-AM assay: (**a**) Irradiated and non-irradiated water-dissolved ZEA samples; (**b**) Irradiated and non-irradiated dried ZEA samples. Bars represent the mean and its standard error (SEM) of at least three independent repetitions. Significant differences with respect to the vehicle control (one-way rmANOVA, Dunnett’s Post-hoc test) are indicated as follows: ** (*p* < 0.01), *** (*p* < 0.001). Statistically significant differences with respect to non-irradiated sample (0 kGy) and the irradiated samples (2.4 kGy and 10.3 kGy) (one-way rmANOVA, Dunnett’s Post-hoc test) are indicated as follows: a (*p* < 0.01), b (*p* < 0.001), no letter (no statistical differences).

**Figure 4 foods-09-01687-f004:**
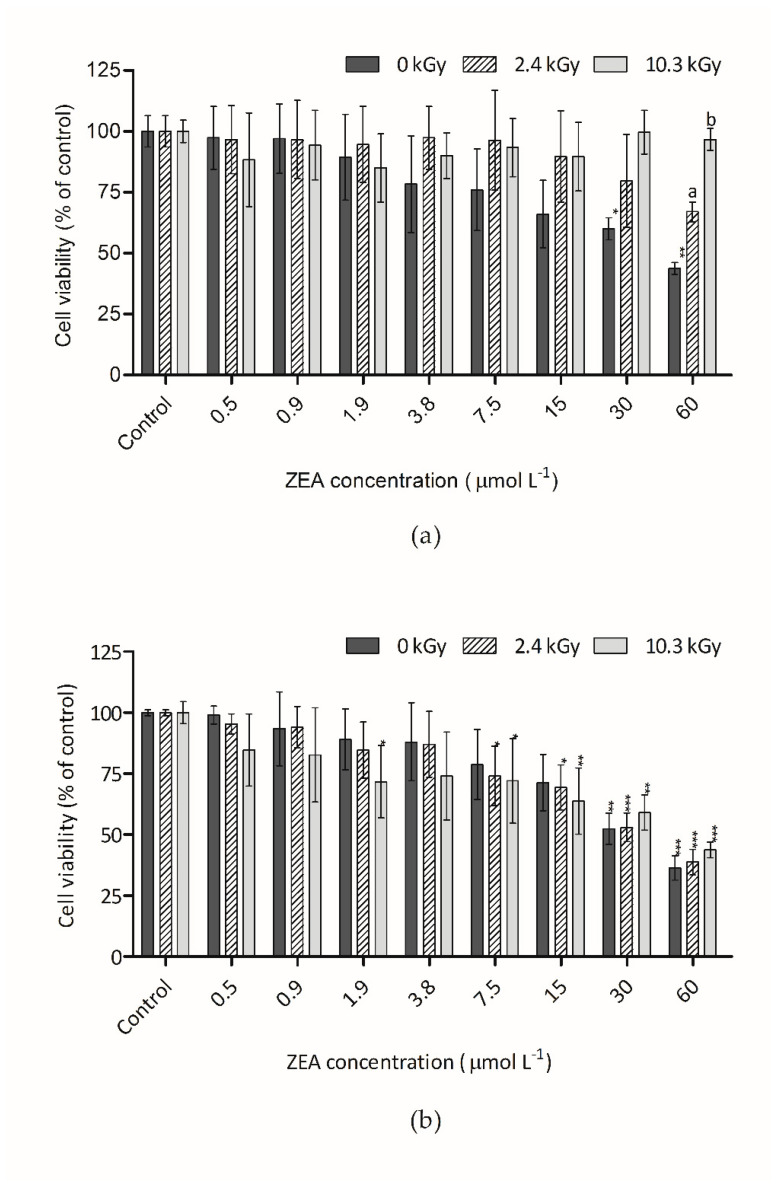
Effect of ZEA samples on HepG2 cell viability assessed by the NR Uptake (NRU) assay: (**a**) Irradiated and non-irradiated water-dissolved ZEA samples; (**b**) Irradiated and non-irradiated dried ZEA samples. Bars represent the mean and its standard error (SEM) of at least three independent repetitions. Significant differences with respect to the vehicle control (one-way rmANOVA, Dunnett’s Post-hoc test) are indicated as follows: * (*p* < 0.05), ** (*p* < 0.01), *** (*p* < 0.001). Statistically significant differences with respect to non-irradiated sample (0 kGy) and the irradiated samples (2.4 kGy and 10.3 kGy) (one-way rmANOVA, Dunnett’s Post-hoc test) are indicated as follows: a (*p* < 0.01), b (*p* < 0.001), no letter (no statistical differences).

**Figure 5 foods-09-01687-f005:**
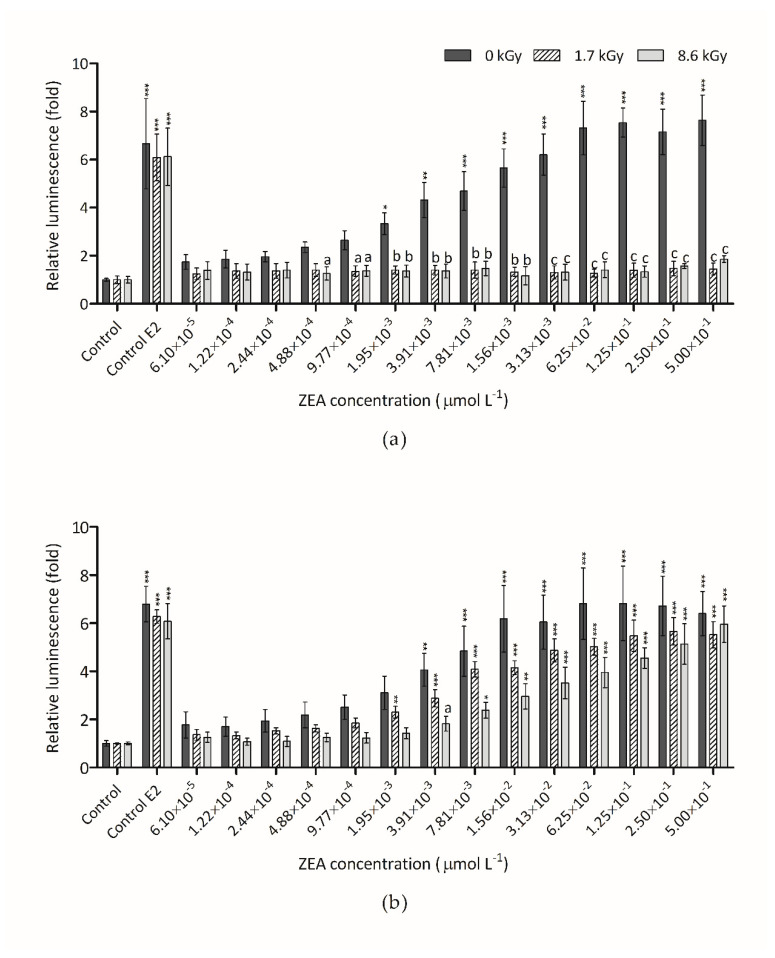
Non-irradiated and irradiated ZEA (3 µmol L^−1^) effect on hERα-HeLa-9903 cells, determined by luciferase activity: (**a**) Water-dissolved ZEA samples; (**b**) Dried ZEA samples. E2 (0.125 nmol L^−1^) was used as positive control. Significant differences with respect to the vehicle control (one-way rmANOVA, Dunnett’s Post-hoc test) are indicated as follows: * (*p* < 0.05), ** (*p* < 0.01), *** (*p* < 0.001). Statistically significant differences with respect to non-irradiated sample (0 kGy) and the irradiated samples (1.7 kGy and 8.6 kGy) (one-way rmANOVA, Dunnett’s Post-hoc test) are indicated as follows: a (*p* < 0.05), b (*p* < 0.01), c (*p* < 0.001), no letter (no statistical differences).

**Figure 6 foods-09-01687-f006:**
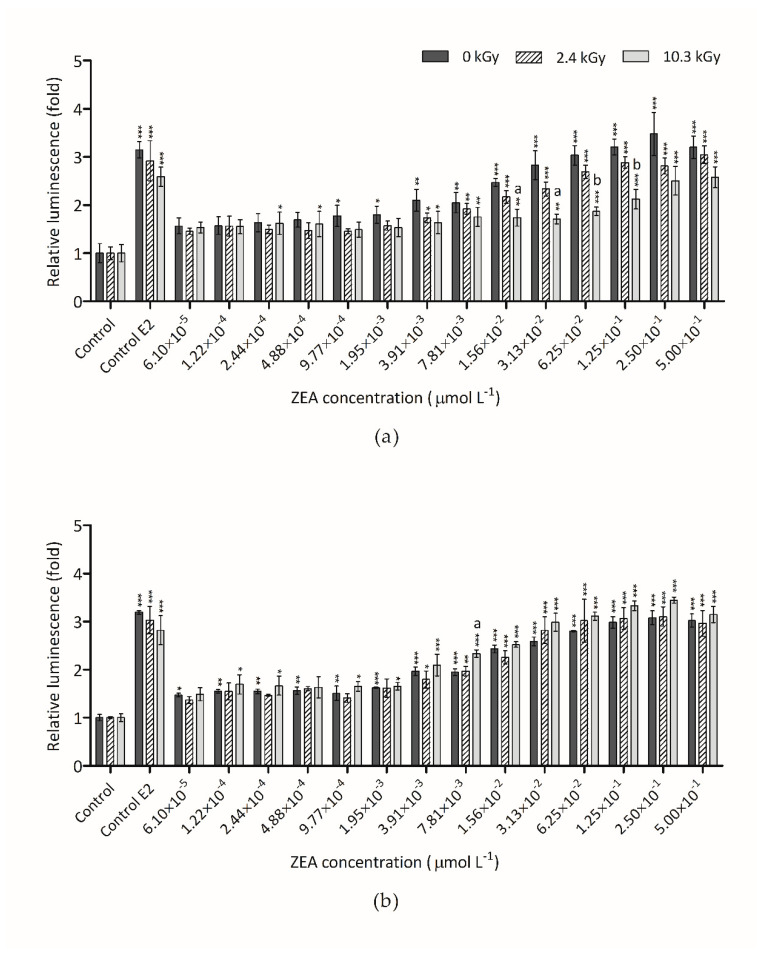
Non-irradiated and irradiated ZEA (60 µmol L^−1^) effect on hERα-HeLa-9903 cells, determined by luciferase activity: (**a**) Water-dissolved ZEA samples; (**b**) Dried ZEA samples. E2 (0.125 nmol L^−1^) was used as positive control. Significant differences with respect to the vehicle control (one-way rmANOVA, Dunnett’s Post-hoc test) are indicated as follows: * (*p* < 0.05), ** (*p* < 0.01), *** (*p* < 0.001). Statistically significant differences with respect to non-irradiated sample (0 kGy) and the irradiated samples (2.4 kGy and 10.3 kGy) (one-way rmANOVA, Dunnett’s Post-hoc test) are indicated as follows: a (*p* < 0.05), b (*p* < 0.01), no letter (no statistical differences).

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
