# Peer review of "Effect of Gamma-Radiation on Zearalenone—Degradation, Cytotoxicity and Estrogenicity"

_foods, 2020, doi:10.3390/foods9111687_

Round 1
Reviewer 1 Report
The article by Calado et al. entitled “Effect of gamma-radiation on Zearalenone – degradation, cytotoxicity and estrogenicity” reports the effect of gamma-radiation on Zearalenone (ZEA) at different moisture conditions, and to evaluate the cytotoxicity and estrogenicity of the irradiated ZEA. The effect of gamma-radiation to eliminate mycotoxins from foods and feeds, is a topic of great interest for researchers and widely investigated as also the same authors affirm. Reduce or eliminate the toxic effects of ZEA is important to improve food safety, and to minimize economic losses in livestock production.
However, novel information on this topic which is more supported (e.g. in vitro assays) is necessary, taking into account also modified mycotoxins as an additional risk to human and animal health. On this basis, the paper deserves after minor changes, as reported below.
Abstract:
-Page 1, line 16: change “foods and feeds” by “food and feed”
Introduction:
-Page 2, line 59: delete “food security”.
-Page 3, line 100: change “gamma radiation” by “gamma-radiation”. Check this in overall text.
Materials and methods section:
-page 3, line 107: authors use “mM” referring to Ultraglutamine 1 molarity. In other cases, (e.g. ZEA samples) authors use “μmol L-1 or nmol L-1”. Please uniform in overall manuscript and when is possible indicate also the concentration as μg mL-1 or ng mL-1 (and so on) reporting this value in round brackets after the molarity value. However, this is not necessary in the figures.
Results and discussion:
-Authors well describe HPLC conditions and related results, but they do not show chromatographic profile to support their data. Please, add a representative chromatogram for each different condition analysed by HPLC where what you claim is visible, such as “the absence of revealed fluorescent degradation products that may have been produced during the irradiation process of ZEA”.
Figures.
-Change “a for p < 0.01, b for p < 0.001” and so on by “a (p < 0.01), b (p < 0.001)”. Uniform this in all figure captions.
-Increase the resolution of Figures, particularly of the writing (e.g. ZEA concentration (% of control)).
-In Figure 1, delete capital letter when referring to “Concentration” on y axis. Moreover, change “gamma radiation” by “gamma-radiation” in figure caption.
Author Response
Dear Reviewer,
We thank you for your valuable comments, and we answer them in the attached file,
Armando

Reviewer 2 Report
In this study, the degradation of mycoestrogen zearalenone was examined by gamma-radiation. The topic is interesting and the manuscript was written in a good scientific style. The experimentation seems to be highly suitable for these investigations. The description of the results is clear and the discussion of the observations is established. I suggest this manuscript need a minor revision. My critical comments are listed below.
Section 3.1: Have the Authors any idea regarding the structure of degradation products formed from ZEA as a result of gamma-radiation? It should be discussed.
Figures: “a”, “b”, “c”, “d” should be defined in the captions
Lines 279-284: These data should be demonstrated as a separate panel in Fig. 1, for the better clarity.
Fig. 2-6: These figs. are too small. Much larger panels should be presented with significantly larger font size, e.g. using upper and lower directions of the panels instead of next to each other.
The level of significance in the Mat&Met should be in agreement with the p values described in fig. captions.
Author Response
Dear Reviewer,
We thank you for your comments, and we provide an answer in the attached file.
Best Regards
Armando

Reviewer 3 Report
Major comments:
- The rationale of irradiating ZEA at the level of 3 micromole per litter (Figure 5) or 60 micromole per litter (Figure 6) is not clear. Besides, the units of the X-axis in Figure 5a and Figure 5b did not match the figure legends of Figure 5.
- To apply the radiation on food juice or cereals, it is important to know if the irradiation would also destroy micro minerals and vitamins. Related information should be added to the discussion.
- Although the author assumes that there is no toxigenic compounds were generated after irradiation (no increase of toxicity was observed), the result did show the quantitively lower cell viabilities on 10.3 kGy groups compared to that in the 0 kGy controls (Figure 4b). It is suggested to add more discussion on these results. Also, the possible structural changes of ZEA after irradiation should be discussed as well.
Minor comments:
- Figure 1. ‘dosis’ => ‘doses’
- Improve the resolutions of Figure 5 and Figure 6.